# Comparing Apples to Oranges: Learning Similarity Functions for Data Produced by Different Distributions

**Leonidas Tsepenekas**
JPMorganChase AI Research
New York, USA
leonidas.tsepenekas@jpmchase.com

**Ivan Brugere**
JPMorganChase AI Research
New York, USA
ivan.brugere@jpmchase.com

**Freddy Lecue**
JPMorganChase AI Research
New York, USA
freddy.lecue@jpmchase.com

**Daniele Maggazeni**
JPMorganChase AI Research
London, UK
daniele.magazzeni@jpmorgan.com

## Abstract

Similarity functions measure how comparable pairs of elements are, and play a key role in a wide variety of applications, e.g., notions of Individual Fairness abiding by the seminal paradigm of Dwork et al. [2012], as well as Clustering problems. However, access to an accurate similarity function should not always be considered guaranteed, and this point was even raised by Dwork et al. [2012]. For instance, it is reasonable to assume that when the elements to be compared are produced by different distributions, or in other words belong to different "demographic" groups, knowledge of their true similarity might be very difficult to obtain. In this work, we present an efficient sampling framework that learns these across-groups similarity functions, using only a limited amount of experts' feedback. We show analytical results with rigorous theoretical bounds, and empirically validate our algorithms via a large suite of experiments.

## 1 Introduction

Given a feature space $\mathcal{I}$, a similarity function $\sigma : \mathcal{I}^2 \mapsto \mathbb{R}_{\geq 0}$ measures how comparable any pair of elements $x, x' \in \mathcal{I}$ are. The function $\sigma$ can also be interpreted as a distance function, where the smaller $\sigma(x, x')$ is, the more similar $x$ and $x'$ are. Such functions are crucially used in a variety of AI/ML problems, *and in each such case $\sigma$ is assumed to be known*.

The most prominent applications where similarity functions have a central role involve considerations of individual fairness. Specifically, all such individual fairness paradigms stem from the the seminal work of Dwork et al. [2012], in which fairness is defined as *treating similar individuals similarly*. In more concrete terms, such paradigms interpret the aforementioned abstract definition of fairness as guaranteeing that for every pair of individuals $x$ and $y$, the difference in the quality of service $x$ and $y$ receive (a.k.a. their received treatment) is upper bounded by their respective similarity value $\sigma(x, y)$; the more similar the individuals are, the less different their quality of service will be. Therefore, any algorithm that needs to abide by such concepts of fairness, should always be able to access the similarity score for every pair of individuals that are of interest.

Another family of applications where similarity functions are vital, involves Clustering problems. In a clustering setting, e.g, the standard $k$-means task, the similarity function is interpreted as a distance

function, that serves as the metric space in which we need to create the appropriate clusters. Clearly, the aforementioned metric space is always assumed to be part of the input.

Nonetheless, it is not realistic to assume that a reliable and accurate similarity function is always given. This issue was even raised in the work of Dwork et al. [2012], where it was acknowledged that the computation of $\sigma$ is not trivial, and thus should be deferred to third parties. The starting point of our work here is the observation that there exist scenarios where computing similarity can be assumed as easy (in other words given), while in other cases this task would be significantly more challenging. Specifically, we are interested in scenarios where there are multiple distributions that produce elements of $\mathcal{I}$. We loosely call each such distribution a "demographic" group, interpreting it as the stochastic way in which members of this group are produced. *In this setting, computing the similarity value of two elements that are produced according to the same distribution, seems intuitively much easier compared to computing similarity values for elements belonging to different groups*. We next present a few motivating examples that clarify this statement.

**Individual Fairness:** Consider a college admissions committee that needs access to an accurate similarity function for students, so that it provides a similar likelihood of acceptance to similar applicants. Let us focus on the following two demographic groups. The first being students from affluent families living in privileged communities and having access to the best quality schools and private tutoring. The other group would consist of students from low-income families, coming from a far less privileged background. Given this setting, the question at hand is **"Should two students with comparable feature vectors (e.g., SAT scores, strength of school curriculum, number of recommendation letters) be really viewed as similar, when they belong to different groups?"** At first, it appears that directly comparing two students based on their features can be an accurate way to elicit their similarity only if the students belong to the same demographic (belonging to the same group serves as a normalization factor). However, this trivial approach might hurt less privileged students when they are compared to students of the first group. This is because such a simplistic way of measuring similarity does not reflect potential that is undeveloped due to unequal access to resources. Hence, accurate across-groups comparisons that take into account such delicate issues, appear considerably more intricate.

**Clustering:** Suppose that a marketing company has a collection of user data that wants to cluster, with its end goal being a downstream market segmentation analysis. However, as it is usually the case, the data might come from different sources, e.g., data from private vendors and data from government bureaus. In this scenario, each data source might have its own way of representing user information, e.g., each source might use a unique subset of features. Therefore, eliciting the distance metric required for the clustering task should be straightforward for data coming from the same source, while across-sources distances would certainly require extra care, e.g., how can one extract the distance of two vectors containing different sets of features?

As suggested by earlier work on computing similarity functions for applications of individual fairness [Ilvento, 2019], when obtaining similarity values is an overwhelming task, one can employ the advice of domain experts. Such experts can be given any pair of elements, and in return produce their true similarity value. However, utilizing this experts' advice can be thought of as very costly, and hence it should be used sparingly. For example, in the case of comparing students from different economic backgrounds, the admissions committee can reach out to regulatory bodies or civil rights organizations. Nonetheless, resorting to these experts for every student comparison that might arise, is clearly not a sustainable solution. Therefore, our goal in this paper is to learn the across-groups similarity functions, using as few queries to experts as possible.

## 2 Preliminaries and contribution

For the ease of exposition, we accompany the formal definitions with brief demonstrations on how they could relate to the previously mentioned college applications use-case.

Let $\mathcal{I}$ denote the feature space of elements; for instance each $x \in \mathcal{I}$ could correspond to a valid student profile. We assume that elements come from $\gamma$ known "demographic" groups, where $\gamma \in \mathbb{N}$, and each group $\ell \in [\gamma]$ is governed by an unknown distribution $\mathcal{D}_\ell$ over $\mathcal{I}$. We use $x \sim \mathcal{D}_\ell$ to denote a randomly drawn $x$ from $\mathcal{D}_\ell$. Further, we use $x \in \mathcal{D}_\ell$ to denote that $x$ is an element in the support of $\mathcal{D}_\ell$, and thus $x$ is a member of group $\ell$. In the college admissions scenario where we have two demographic groups, there will be two distributions $\mathcal{D}_1$ and $\mathcal{D}_2$ dictating how the profiles of

privileged and non-privileged students are respectively produced. Observe now that for a specific $x \in \mathcal{I}$, we might have $x \in \mathcal{D}_\ell$ and $x \in \mathcal{D}_{\ell'}$, for $\ell \neq \ell'$. Hence, in our model group membership is important, and every time we are considering an element $x \in \mathcal{I}$, we know which distribution produced $x$, e.g., whether the profile $x$ belongs to a privileged or non-privileged student.

For every group $\ell \in [\gamma]$ there is an intra-group similarity function $d_\ell : \mathcal{I}^2 \mapsto \mathbb{R}_{\geq 0}$, such that for all $x, y \in \mathcal{D}_\ell$ we have $d_\ell(x, y)$ representing the true similarity between $x, y$. In addition, the smaller $d_\ell(x, y)$ is, the more similar $x, y$. Note here that the function $d_\ell$ is only used to compare members of group $\ell$ (in the college admissions example, the function $d_1$ would only be used to compare privileged students with each other). Further, a common assumption for functions measuring similarity is that they are metric[1] [Yona and Rothblum, 2018, Kim et al., 2018, Ilvento, 2019, Mukherjee et al., 2020, Wang et al., 2019]. *We also adopt the metric assumption for the function $d_\ell$.* Finally, based on the earlier discussion regarding computing similarity between elements of the same group, we assume that $d_\ell$ is known, and given as part of the instance.

Moreover, for any two groups $\ell$ and $\ell'$ there exists an *unknown* across-groups similarity function $\sigma_{\ell,\ell'} : \mathcal{I}^2 \mapsto \mathbb{R}_{\geq 0}$, such that for all $x \in \mathcal{D}_\ell$ and $y \in \mathcal{D}_{\ell'}$, $\sigma_{\ell,\ell'}(x, y)$ represents the true similarity between $x, y$. Again, the smaller $\sigma_{\ell,\ell'}(x, y)$ is, the more similar the two elements, and for a meaningful use of $\sigma_{\ell,\ell'}$ we must make sure that $x$ is a member of group $\ell$ and $y$ a member of group $\ell'$. In the college admissions scenario, $\sigma_{1,2}$ is the way you can accurately compare a privileged and a non-privilaged student. Finally, to capture the metric nature of a similarity function, we impose the following mild properties on $\sigma_{\ell,\ell'}$, which can be viewed as across-groups triangle inequalities:

1. **Property $\mathcal{M}_1$:** $\sigma_{\ell,\ell'}(x, y) \leq d_\ell(x, z) + \sigma_{\ell,\ell'}(z, y)$ for every $x, z \in \mathcal{D}_\ell$ and $y \in \mathcal{D}_{\ell'}$.
2. **Property $\mathcal{M}_2$:** $\sigma_{\ell,\ell'}(x, y) \leq \sigma_{\ell,\ell'}(x, z) + d_{\ell'}(z, y)$ for every $x \in \mathcal{D}_\ell$ and $y, z \in \mathcal{D}_{\ell'}$.

In terms of the college admissions use-case, $\mathcal{M}_1$ and $\mathcal{M}_2$ try to capture reasonable assumptions of the following form. If a non-privileged student $x$ is similar to another non-privileged student $z$, and $z$ is similar to a privileged student $y$, then $x$ and $y$ should also be similar to each other.

*Observe now that the collection of all similarity values (intra-group and across-groups) in our model does not axiomatically yield a valid metric space.* This is due to the following reasons. **1)** If $\sigma_{\ell,\ell'}(x, y) = 0$ for $x \in \mathcal{D}_\ell$ and $y \in \mathcal{D}_{\ell'}$, then we do not necessarily have $x = y$. **2)** It is not always the case that $d_\ell(x, y) \leq \sigma_{\ell,\ell'}(x, z) + \sigma_{\ell,\ell'}(y, z)$ for $x, y \in \mathcal{D}_\ell, z \in \mathcal{D}_{\ell'}$. **3)** It is not always the case that $\sigma_{\ell,\ell'}(x, y) \leq \sigma_{\ell,\ell''}(x, z) + \sigma_{\ell'',\ell'}(z, y)$ for $x \in \mathcal{D}_\ell, y \in \mathcal{D}_{\ell'}, z \in \mathcal{D}_{\ell''}$.

However, not having the collection of similarity values necessarily produce a metric space is not a weakness of our model. On the contrary, we view this as one of its strongest aspects. For one thing, imposing a complete metric constraint on the case of intricate across-groups comparisons sounds unrealistic and very restrictive. Further, even though existing literature treats similarity functions as metric ones, the seminal work of Dwork et al. [2012] mentions that this should not always be the case. Hence, our model is more general than the current literature.

**Goal of Our Problem:** We want for any two groups $\ell, \ell'$ to compute a function $f_{\ell,\ell'} : \mathcal{I}^2 \mapsto \mathbb{R}_{\geq 0}$, such that $f_{\ell,\ell'}(x, y)$ is our estimate of similarity for any $x \in \mathcal{D}_\ell$ and $y \in \mathcal{D}_{\ell'}$. Specifically, we seek a PAC (Probably Approximately Correct) guarantee, where for any given accuracy and confidence parameters $\epsilon, \delta \in (0, 1)$ we have:

$$\Pr_{x \sim \mathcal{D}_\ell, y \sim \mathcal{D}_{\ell'}} \left[ \left| f_{\ell,\ell'}(x, y) - \sigma_{\ell,\ell'}(x, y) \right| > \epsilon \right] \leq \delta$$

The subscript in the above probability corresponds to two independent random choices, one $x \sim \mathcal{D}_\ell$ and one $y \sim \mathcal{D}_{\ell'}$. In other words, we want for any given pair our estimate to be $\epsilon$-close to the real similarity value, with probability at least $1 - \delta$, where $\epsilon$ and $\delta$ are user-specified parameters.

As for tools to learn $f_{\ell,\ell'}$, we only require two things. At first, for each group $\ell$ we want a set $S_\ell$ of i.i.d. samples from $\mathcal{D}_\ell$. Obviously, the total number of used samples should be polynomial in the input parameters, i.e., polynomial in $\gamma$, $\frac{1}{\epsilon}$ and $\frac{1}{\delta}$. Secondly, we require access to an expert oracle, which given any $x \in S_\ell$ and $y \in S_{\ell'}$ for any $\ell$ and $\ell'$, returns the true similarity value $\sigma_{\ell,\ell'}(x, y)$. We refer to a single invocation of the oracle as a query. Since there is a cost to collecting expert feedback, an additional objective in our problem is minimizing the number of oracle queries.

---

[1] A function $d$ is metric if **a)** $d(x, y) = 0$ iff $x = y$, **b)** $d(x, y) = d(y, x)$ and **c)** $d(x, y) \leq d(x, z) + d(z, y)$ for all $x, y, z$ (triangle inequality).

## 2.1 Outline and discussion of our results

In Section 4 we present our theoretical results. We begin with a simple and very intuitive learning algorithm which achieves the following guarantees.

**Theorem 2.1.** *For any given parameters $\epsilon, \delta \in (0, 1)$, the simple algorithm produces a similarity approximation function $f_{\ell, \ell'}$ for every $\ell$ and $\ell'$, such that:*

$$\Pr[Error_{(\ell, \ell')}] := \Pr_{\substack{x \sim \mathcal{D}_\ell, \\ y \sim \mathcal{D}_{\ell'}, \, \mathcal{A}}} \left[ \left| f_{\ell, \ell'}(x, y) - \sigma_{\ell, \ell'}(x, y) \right| = \omega(\epsilon) \right]$$

$$= O\left( \delta + p_\ell(\epsilon, \delta) + p_{\ell'}(\epsilon, \delta) \right)$$

*The randomness here is of three independent sources. The internal randomness $\mathcal{A}$ of the algorithm, a choice $x \sim \mathcal{D}_\ell$, and a choice $y \sim \mathcal{D}_{\ell'}$. The algorithm requires $\frac{1}{\delta} \log \frac{1}{\delta^2}$ samples from each group, and utilizes $\frac{\gamma(\gamma-1)}{\delta^2} \log^2 \frac{1}{\delta^2}$ oracle queries.*

In plain English, the above theorem says that with a polynomial number of samples and queries, the algorithm achieves an $O(\epsilon)$ accuracy with high probability, i.e., with probability $\omega\left(1 - \delta - p_\ell(\epsilon, \delta) - p_{\ell'}(\epsilon, \delta)\right)$. The definition of the functions $p_\ell$ is presented next.

**Definition 2.2.** For each group $\ell$, we use $p_\ell(\epsilon, \delta)$ to denote the probability of sampling an $(\epsilon, \delta)$-rare element of $\mathcal{D}_\ell$. We define as $(\epsilon, \delta)$-rare for $\mathcal{D}_\ell$, an element $x \in \mathcal{D}_\ell$ for which there is a less than $\delta$ chance of sampling $x' \sim \mathcal{D}_\ell$ with $d_\ell(x, x') \leq \epsilon$. Formally, $x \in \mathcal{D}_\ell$ is $(\epsilon, \delta)$-rare iff $\Pr_{x' \sim \mathcal{D}_\ell}[d_\ell(x, x') \leq \epsilon] < \delta$, and $p_\ell(\epsilon, \delta) = \Pr_{x \sim \mathcal{D}_\ell}[x \text{ is } (\epsilon, \delta)\text{-rare for } \mathcal{D}_\ell]$. Intuitively, a rare element should be interpreted as an "isolated" member of the group, in the sense that it is at most $\delta$-likely to encounter another element that is $\epsilon$-similar to it. For instance, a privileged student is considered isolated, if only a small fraction of other privileged students have a profile similar to them.

Clearly, to get a PAC guarantee where the algorithm's error probability for $\ell$ and $\ell'$ is $O(\delta)$, we need $p_\ell(\epsilon, \delta), p_{\ell'}(\epsilon, \delta) = O(\delta)$. We hypothesize that in realistic distributions each $p_\ell(\epsilon, \delta)$ should indeed be fairly small, and this hypothesis is actually validated by our experiments. The reason we believe this hypothesis to be true, is that very frequently real data demonstrate high concentration around certain archetypal elements. Hence, this sort of distributional density does not leave room for isolated elements in the rest of the space. Nonetheless, we also provide a strong *no free lunch* result for the values $p_\ell(\epsilon, \delta)$, which shows that any practical PAC-algorithm necessarily depends on them. *This result further implies that our algorithm's error probabilities are indeed almost optimal.*

**Theorem 2.3** (No-Free Lunch Theorem)**.** *For any given $\epsilon, \delta \in (0, 1)$, any algorithm using finitely many samples, will yield similarity approximations $f_{\ell, \ell'}$ with $\Pr\left[|f_{\ell, \ell'}(x, y) - \sigma_{\ell, \ell'}(x, y)| = \omega(\epsilon)\right] = \Omega(\max\{p_\ell(\epsilon, \delta), p_{\ell'}(\epsilon, \delta)\} - \epsilon)$; the probability is over the independent choices $x \sim \mathcal{D}_\ell$ and $y \sim \mathcal{D}'_\ell$ as well as any potential internal randomness of the algorithm.*

In plain English, Theorem 2.3 says that any algorithm using a finite amount of samples, can achieve $\epsilon$-accuracy with a probability that is necessarily at most $1 - \max\{p_\ell(\epsilon, \delta), p_{\ell'}(\epsilon, \delta)\} + \epsilon$, i.e., if $\max\{p_\ell(\epsilon, \delta), p_{\ell'}(\epsilon, \delta)\}$ is large, learning is impossible.

Moving on, we focus on minimizing the oracle queries. By carefully modifying the earlier simple algorithm, we obtain a new more intricate algorithm with the following guarantees:

**Theorem 2.4.** *For any given parameters $\epsilon, \delta \in (0, 1)$, the query-minimizing algorithm produces similarity approximation functions $f_{\ell, \ell'}$ for every $\ell$ and $\ell'$, such that:*

$$\Pr[Error_{(\ell, \ell')}] = O(\delta + p_\ell(\epsilon, \delta) + p_{\ell'}(\epsilon, \delta))$$

$\Pr[Error_{(\ell, \ell')}]$ *is as defined in Theorem 2.1. Let $N = \frac{1}{\delta} \log \frac{1}{\delta^2}$. The algorithm requires $N$ samples from each group, and the number of oracle queries used is at most*

$$\sum_{\ell \in [\gamma]} \left( Q_\ell \sum_{\ell' \in [\gamma]: \, \ell' \neq \ell} Q_{\ell'} \right)$$

*where $Q_\ell \leq N$ and $\mathbb{E}[Q_\ell] \leq \frac{1}{\delta} + p_\ell(\epsilon, \delta) N$ for each $\ell$.*

At first, the confidence, accuracy and sample complexity guarantees of the new algorithm are the same as those of the simpler one described in Theorem 2.1. Furthermore, because $Q_\ell \leq N$, the

queries of the improved algorithm are at most $\gamma(\gamma - 1)N$, which is exactly the number of queries in our earlier simple algorithm. However, the smaller the values $p_\ell(\epsilon, \delta)$ are, the fewer queries in expectation. Our experimental results indeed confirm that the improved algorithm always leads to a significant decrease in the used queries.

Our final theoretical result involves a lower bound on the number of queries required for learning.

**Theorem 2.5.** *For all $\epsilon, \delta \in (0, 1)$, any learning algorithm producing similarity approximation functions $f_{\ell, \ell'}$ with $\mathrm{Pr}_{x \sim \mathcal{D}_\ell, y \sim \mathcal{D}_{\ell'}} \left[ |f_{\ell, \ell'}(x, y) - \sigma_{\ell, \ell'}(x, y)| = \omega(\epsilon) \right] = O(\delta)$, needs $\Omega(\frac{\gamma^2}{\delta^2})$ queries.*

Combining Theorems 2.4 and 2.5 implies that when all $p_\ell(\epsilon, \delta)$ are negligible, i.e., $p_\ell(\epsilon, \delta) \to 0$, *the expected queries of the Theorem 2.4 algorithm are asymptotically optimal.*

Finally, Section 5 contains our experimental evaluation, where through a large suite of simulations we validate our theoretical findings.

# 3 Related work

Metric learning is a very well-studied area [Bellet et al., 2013, Kulis, 2013, Moutafis et al., 2017, Suárez-Díaz et al., 2018]. There is also an extensive amount of work on using human feedback for learning metrics in specific tasks, e.g., image similarity and low-dimensional embeddings [Frome et al., 2007, Jamieson and Nowak, 2011, Tamuz et al., 2011, van der Maaten and Weinberger, 2012, Wilber et al., 2014]. However, since these works are either tied to specific applications or specific metrics, they are only distantly related to ours.

Our model is more closely related to the literature on trying to learn the similarity function from the fairness definition of Dwork et al. [2012]. This concept of fairness requires treating similar individuals similarly. Thus, it needs access to a function that returns a non-negative value for any pair of individuals, and this value corresponds to how similar the individuals are. Specifically, the smaller the value the more similar the elements that are compared. Even though the fairness definition of Dwork et al. [2012] is very elegant and intuitive, the main obstacle for adopting it in practice is the inability to easily compute or access the crucial similarity function. To our knowledge, the only papers that attempt to learn this similarity function using expert oracles like us, are Ilvento [2019], Mukherjee et al. [2020] and Wang et al. [2019]. Ilvento [2019] addresses the scenario of learning a general metric function, and gives theoretical PAC guarantees. Mukherjee et al. [2020] give theoretical guarantees for learning similarity functions that are only of a specific Mahalanobis form. Wang et al. [2019] simply provide empirical results. The first difference between our model and these papers is that unlike us, they do not consider elements coming from multiple distributions. However, the most important difference is that these works only learn **metric** functions. **In our case the collection of similarity values (from all $d_\ell$ and $\sigma_{\ell, \ell'}$) does not necessarily yield a complete metric space**; see the discussion in Section 2. Hence, our problem addresses more general functions.

Regarding the difficulty in computing similarity between members of different groups, we are only aware of a brief result by Dwork et al. [2012]. In particular, given a metric $d$ over the whole feature space, they mention that $d$ can only be trusted for comparisons between elements of the same group, and not for across-groups comparisons. In order to achieve the latter for groups $\ell$ and $\ell'$, they find a new similarity function $d'$ that approximates $d$, while minimizing the Earthmover distance between the distributions $\mathcal{D}_\ell, \mathcal{D}_{\ell'}$. This is completely different from our work, since here we assume the existence of across-groups similarity values, which we eventually want to learn. On the other hand, the approach of Dwork et al. [2012] can be seen as an optimization problem, where the across-groups similarity values need to be computed in a way that minimizes some objective. Also, unlike our model, this optimization approach has a serious limitation, and that is requiring $\mathcal{D}_\ell, \mathcal{D}_{\ell'}$ to be explicitly known (recall that here we only need samples from these distributions).

Finally, since similarity as distance is difficult to compute in practice, there has been a line of research that defines similarity using simpler, yet less expressive structures. Examples include similarity lists Chakrabarti et al. [2022], graphs Lahoti et al. [2019] and ordinal relationships Jung et al. [2019].

# 4 Theoretical results

All proofs for this section can be found in the supplementary material

## 4.1 A simple learning algorithm

Given any confidence and accuracy parameters $\delta, \epsilon \in (0, 1)$ respectively, our approach is summarized as follows. At first, for every group $\ell$ we need a set $S_\ell$ of samples that are chosen i.i.d. according to $\mathcal{D}_\ell$, such that $|S_\ell| = \frac{1}{\delta} \log \frac{1}{\delta^2}$. Then, for every distinct $\ell$ and $\ell'$, and for all $x \in S_\ell$ and $y \in S_{\ell'}$, we ask the expert oracle for the true similarity value $\sigma_{\ell, \ell'}(x, y)$. The next observation follows trivially.

**Observation 4.1.** The algorithm uses $\frac{\gamma}{\delta} \log \frac{1}{\delta^2}$ samples, and $\frac{\gamma(\gamma-1)}{\delta^2} \log^2 \frac{1}{\delta^2}$ queries to the oracle.

Suppose now that we need to compare any $x \in \mathcal{D}_\ell$ and $y \in \mathcal{D}_{\ell'}$. Our high level idea is that the properties $\mathcal{M}_1$ and $\mathcal{M}_2$ of $\sigma_{\ell, \ell'}$ (see Section 2), will actually allow us to use the closest element to $x$ in $S_\ell$ and the closest element to $y$ in $S_{\ell'}$ as proxies. Thus, let $\pi(x) = \arg \min_{x' \in S_\ell} d_\ell(x, x')$ and $\pi(y) = \arg \min_{y' \in S_{\ell'}} d_{\ell'}(y, y')$. The algorithm then sets

$$f_{\ell, \ell'}(x, y) := \sigma_{\ell, \ell'}(\pi(x), \pi(y))$$

where $\sigma_{\ell, \ell'}(\pi(x), \pi(y))$ is known from the earlier queries.

Before we proceed with the analysis of the algorithm, we need to recall some notation which was introduced in Section 2.1. Consider any group $\ell$. An element $x \in \mathcal{D}_\ell$ with $\Pr_{x' \sim \mathcal{D}_\ell}[d_\ell(x, x') \leq \epsilon] < \delta$ is called an $(\epsilon, \delta)$-rare element of $\mathcal{D}_\ell$, and also $p_\ell(\epsilon, \delta) := \Pr_{x \sim \mathcal{D}_\ell}[x \text{ is } (\epsilon, \delta)\text{-rare for } \mathcal{D}_\ell]$.

**Theorem 4.2.** *For any given parameters $\epsilon, \delta \in (0, 1)$, the simple algorithm produces similarity approximation functions $f_{\ell, \ell'}$ for every $\ell$ and $\ell'$, such that*

$$\Pr[Error_{(\ell, \ell')}] = O(\delta + p_\ell(\epsilon, \delta) + p_{\ell'}(\epsilon, \delta))$$

*where $\Pr[Error_{(\ell, \ell')}]$ is as in Theorem 2.1.*

Observation 4.1 and Theorem 4.2 directly yield Theorem 2.1.

A potential criticism of the algorithm presented here, is that its error probabilities depend on $p_\ell(\epsilon, \delta)$. However, Theorem 2.3 shows that such a dependence is unavoidable (see appendix for proof).

## 4.2 Optimizing the number of expert queries

Here we modify the earlier algorithm in a way that improves the number of queries used. The idea behind this improvement is the following. Given the sets of samples $S_\ell$, instead of asking the oracle for all possible similarity values $\sigma_{\ell, \ell'}(x, y)$ for every $\ell, \ell'$ and every $x \in S_\ell$ and $y \in S_{\ell'}$, we would rather choose a set $R_\ell \subseteq S_\ell$ of representative elements for each group $\ell$. Then, we would ask the oracle for the values $\sigma_{\ell, \ell'}(x, y)$ for every $\ell, \ell'$, but this time only for every $x \in R_\ell$ and $y \in R_{\ell'}$. The choice of the representatives is inspired by the $k$-center algorithm of Hochbaum and Shmoys [1985]. Intuitively, the representatives $R_\ell$ of group $\ell$ will serve as similarity proxies for the elements of $S_\ell$, such that each $x \in S_\ell$ is assigned to a nearby $r_\ell(x) \in R_\ell$ via a mapping function $r_\ell : S_\ell \mapsto R_\ell$. Hence, if $d_\ell(x, r_\ell(x))$ is small enough, $x$ and $r_\ell(x)$ are highly similar, and thus $r_\ell(x)$ acts as a good approximation of $x$. The full details for the construction of $R_\ell, r_\ell$ are presented in Algorithm 1.

Suppose now that we need to compare some $x \in \mathcal{D}_\ell$ and $y \in \mathcal{D}_{\ell'}$. Our approach will be almost identical to that of Section 4.1. Once again, let $\pi(x) = \arg \min_{x' \in S_\ell} d_\ell(x, x')$ and $\pi(y) = \arg \min_{y' \in S_{\ell'}} d_{\ell'}(y, y')$. However, unlike the simple algorithm of Section 4.1 that directly uses $\pi(x)$ and $\pi(y)$, the more intricate algorithm here will rather use their proxies $r_\ell(\pi(x))$ and $r_{\ell'}(\pi(y))$. Our prediction will then be

$$f_{\ell, \ell'}(x, y) := \sigma_{\ell, \ell'}\Big(r_\ell\big(\pi(x)\big), r_{\ell'}\big(\pi(y)\big)\Big)$$

where $\sigma_{\ell, \ell'}(r_\ell(\pi(x)), r_{\ell'}(\pi(y)))$ is known from the earlier queries.

**Theorem 4.3.** *For any given parameters $\epsilon, \delta \in (0, 1)$, the new query optimization algorithm produces similarity approximation functions $f_{\ell, \ell'}$ for every $\ell$ and $\ell'$, such that*

$$\Pr[Error_{(\ell, \ell')}] = O(\delta + p_\ell(\epsilon, \delta) + p_{\ell'}(\epsilon, \delta))$$

*where $\Pr[Error_{(\ell, \ell')}]$ is as in Theorem 2.1.*

Since the number of samples used by the algorithm is easily seen to be $\frac{\gamma}{\delta} \log \frac{1}{\delta^2}$, the only thing left in order to prove Theorem 2.4 is analyzing the number of oracle queries. To that end, for every group $\ell \in [\gamma]$ with its sampled set $S_\ell$, we define the following Set Cover problem.

---

**Algorithm 1** Training Phase

---

Accuracy and confidence parameters $\epsilon, \delta$. For every group $\ell \in [\gamma]$, a set $S_\ell$ of i.i.d. samples chosen according to $\mathcal{D}_\ell$, such that $|S_\ell| = \frac{1}{\delta} \log \frac{1}{\delta^2}$.

1: **for** each $\ell \in [\gamma]$ **do**
2:     $H_x^\ell \leftarrow \{x' \in S_\ell : d_\ell(x, x') \leq 8\epsilon\}$ for each $x \in S_\ell$.
3:     $U \leftarrow S_\ell, R_\ell \leftarrow \emptyset$ and $r_\ell(x) \leftarrow x$ for each $x \in S_\ell$.
4:     **while** $U \neq \emptyset$ **do**
5:        Choose an arbitrary $x \in U$.
6:        Set $R_\ell \leftarrow R_\ell \cup \{x\}$.
7:        $W_x \leftarrow \{x' \in U : H_x^\ell \cap H_{x'}^\ell \neq \emptyset\}$.
8:        $r_\ell(x') \leftarrow x$ for every $x' \in W_x$.
9:        $U \leftarrow U \setminus W_x$.
10:    **end while**
11: **end for**
12: For every distinct $\ell$ and $\ell'$, and for every $x \in R_\ell$ and $y \in R_{\ell'}$, ask the oracle for $\sigma_{\ell,\ell'}(x, y)$ and store all these values.
13: For every $\ell$, return the set $R_\ell$ and the function $r_\ell$.

---

**Definition 4.4.** Let $\mathcal{H}_x^\ell := \{x' \in S_\ell : d_\ell(x, x') \leq 4\epsilon\}$ for all $x \in S_\ell$. Find $C \subseteq S_\ell$ minimizing $|C|$, with $\bigcup_{c \in C} \mathcal{H}_c^\ell = S_\ell$. We use $OPT_\ell$ to denote the optimal value of this problem. Using standard terminology, we say $x \in S_\ell$ is *covered* by $C$ if $x \in \bigcup_{c \in C} \mathcal{H}_c^\ell$, and $C$ is *feasible* if it covers all $x \in S_\ell$.

**Lemma 4.5.** *For every $\ell \in [\gamma]$ we have $|R_\ell| \leq OPT_\ell$.*

**Lemma 4.6.** *Let $N = \frac{1}{\delta} \log \frac{1}{\delta^2}$. For each group $\ell \in [\gamma]$ we have $OPT_\ell \leq N$ with probability $1$, and $\mathbb{E}[OPT_\ell] \leq \frac{1}{\delta} + p_\ell(\epsilon, \delta)N$. The randomness here is over the samples $S_\ell$.*

The proof of Theorem 2.4 follows since all pairwise queries for the elements in the sets $R_\ell$ are

$$\sum_\ell \left( |R_\ell| \sum_{\ell' \neq \ell} |R_{\ell'}| \right) \leq \sum_\ell \left( OPT_\ell \sum_{\ell' \neq \ell} OPT_{\ell'} \right) \tag{1}$$

where the inequality follows from Lemma 4.5. Combining (1) and Lemma 4.6 proves Theorem 2.4.

**Remark 4.7.** *The factor $8$ in the definition of $H_x^\ell$ at line 2 of Algorithm 1 is arbitrary. Actually, any factor $\rho = O(1)$ would yield the same asymptotic guarantees, with any changes in accuracy and queries being only of an $O(1)$ order of magnitude. Specifically, the smaller $\rho$ is, the better the achieved accuracy and the more queries we are using.*

Finally, we are interested in lower bounds on the queries required for learning. Thus, we give Theorem 2.5, showing that any algorithm with accuracy $O(\epsilon)$ and confidence $O(\delta)$ needs $\Omega(\gamma^2/\delta^2)$ queries. This immediately leads to the following corollary characterizing the optimality of our algorithm.

**Corollary 4.8.** *When for every $\ell \in [\gamma]$ the value $p_\ell(\epsilon, \delta)$ is arbitrarily close to $0$, the algorithm presented in this section achieves an expected number of oracle queries that is asymptotically optimal.*

## 5 Experimental evaluation

We implemented all algorithms in Python 3.10.6 and ran our experiments on a personal laptop with Intel(R) Core(TM) i7-7500U CPU @ 2.70GHz 2.90 GHz and 16.0 GB memory.

**Algorithms:** We implemented the simple algorithm from Section 4.1, and the more intricate algorithm of Section 4.2. We refer to the former as NAIVE, and to the latter as CLUSTER. For the training phase of CLUSTER, we set the dilation factor at line 2 of Algorithm 1 to 2 instead of 8. The reason for this, is that a minimal experimental investigation revealed that this choice leads to a good balance between accuracy guarantees and oracle queries. As explained in Section 3, neither the existing similarity learning algorithms [Ilvento, 2019, Mukherjee et al., 2020, Wang et al., 2019] nor the Earthmover minimization approach of Dwork et al. [2012] address the problem of finding similarity values for heterogeneous data. Furthermore, if the across-groups functions $\sigma$ are not metric, then no approach from the metric learning literature can be used. Hence, as baselines we used three general regression

models; an MLP, a Random Forest regressor (RF) and an XGBoost regressor (XGB). The MLP uses 4 hidden layers of 32 relu activation nodes, and both RF and XGB use 200 estimators.

**Number of demographic groups:** All our experiments are performed for two groups, i.e., $\gamma = 2$. The following reasons justify this decision. At first, this case captures the essence of our algorithmic results; the $\gamma > 2$ case can be viewed as running the algorithm for $\gamma = 2$ multiple times, one for each pair of groups. Secondly, as the theoretical guarantees suggest, the achieved confidence and accuracy of our algorithms are completely independent of $\gamma$.

**Similarity functions:** In all our experiments the feature space is $\mathbb{R}^d$, where $d \in \mathbb{N}$ is case-specific. In line with our motivation which assumes that the intra-group similarity functions are simple, we define $d_1$ and $d_2$ to be the Euclidean distance. Specifically, for $\ell \in \{1, 2\}$, the similarity between any $x, y \in \mathcal{D}_\ell$ is given by

$$d_\ell(x, y) = \sqrt{\sum_{i \in [d]} (x_i - y_i)^2}$$

For the across-groups similarities, we aim for a difficult to learn *non-metric* function; we purposefully chose a non-trivial function in order to challenge both our algorithms and the baselines. Namely, for any $x \in \mathcal{D}_1$ and $y \in \mathcal{D}_2$, we assume

$$\sigma(x, y) = \sqrt[3]{\sum_{i \in [d]} |\alpha_i \cdot x_i - \beta_i \cdot y_i + \theta_i|^3} \tag{2}$$

where the vectors $\alpha, \beta, \theta \in \mathbb{R}^d$ are basically the hidden parameters to be learned (of course non of the learners we use has any insight on the specific structure of $\sigma$).

The function $\sigma$ implicitly adopts a paradigm of feature importance [Niño-Adan et al., 2021]. Specifically, when $x$ and $y$ are to be compared, their features are scaled accordingly by the expert using the parameters $\alpha, \beta$, while some offsetting via $\theta$ might also be necessary. For example, in the college admissions use-case, certain features may have to be properly adjusted (increase a feature for a non-privileged student and decrease it for the privileged one). In the end, the similarity is calculated in an $\ell_3$-like manner. To see why $\sigma$ is not a metric and why it satisfies the necessary properties $\mathcal{M}_1$ and $\mathcal{M}_2$ from Section 2, refer to the last theorem in the Appendix.

In each experiment we choose all $\alpha_i, \beta_i, \theta_i$ independently. The values $\alpha_i, \beta_i$ are chosen uniformly at random from $[0, 1]$, while the $\theta_i$ are chosen uniformly at random from $[-0.01, 0.01]$. Obviously, the algorithms do not have access to the vectors $\alpha, \beta, \theta$, which are only used to simulate the oracle and compare our predictions with the corresponding true values.

**Datasets:** We used 2 datasets from the UCI ML Repository, namely Adult (48,842 points - 14 features) [Kohavi, 1996] and Credit Card Default (30,000 points - 23 features) [Yeh and Lien, 2009], and the publicly available Give Me Some Credit dataset (150,000 points - 11 features) [Credit Fusion, 2011]. We chose these datasets because this type of data is frequently used in applications of issuing credit scores, and in such cases fairness considerations are of utmost importance. For Adult, where categorical features are not encoded as integers, we assigned each category to an integer in $\{1, \#categories\}$, and this integer is used in place of the category in the feature vector [Ding et al., 2021]. Finally, in every dataset we standardized all features through a MinMax re-scaller. Due to space constraints, the figures for Adult and Give me Some Credit are moved to the supplementary material. However, the observed traits there the same as the ones shown here for Credit Card Default.

**Choosing the two groups:** For Credit Card Default and Adult, we defined groups based on marital status. Specifically, the first group corresponds to points that are married individuals, and the second to points that are not married (singles, divorced and widowed are merged together). In Give Me Some Credit, we partition individuals into two groups based on whether or not they have dependents.

**Choosing the accuracy parameter $\epsilon$:** To use a meaningful value for $\epsilon$, we need to know the order of magnitude of $\sigma(x, y)$. Thus, we calculated the value of $\sigma(x, y)$ over $10,000$ trials, where the randomness was of multiple factors, i.e., the random choices for the $\alpha, \beta, \theta$, and the sampling of $x$ and $y$. Figure 1 shows histograms for the empirical frequency of $\sigma(x, y)$ over the $10,000$ runs. In addition, in those trials the minimum value of $\sigma(x, y)$ observed was **1)** 0.1149 for Credit Card Default, **2)** 0.1155 for Adult, and **3)** 0.0132 for Give Me Some Credit. Thus, aiming for an accuracy parameter that is at least an order of magnitude smaller than the value to be learned, we choose $\epsilon = 0.01$ for Credit Card Default and Adult, and $\epsilon = 0.001$ for Give Me Some Credit.

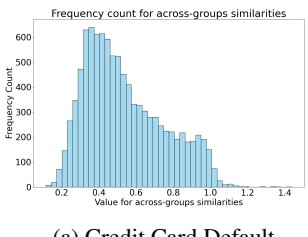

(a) Credit Card Default

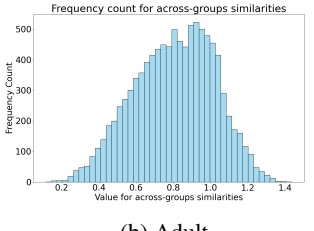

(b) Adult

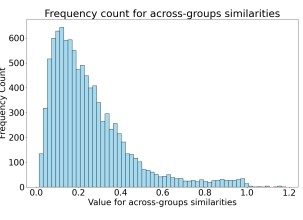

(c) Give Me Some Credit

Figure 1: Frequency counts for $\sigma(x, y)$

Table 1: Error statistics for Credit Card Default

| Algorithm | Average Relative Error % | SD of Relative Error % | Average (Absolute Error)/$\epsilon$ | SD of (Absolute Error)/$\epsilon$ |
|---|---|---|---|---|
| NAIVE | 1.558 | 3.410 | 0.636 | 1.633 |
| CLUSTER | 1.593 | 3.391 | 0.646 | 1.626 |
| MLP | 3.671 | 4.275 | 1.465 | 1.599 |
| RF | 3.237 | 4.813 | 1.306 | 2.318 |
| XGB | 3.121 | 4.606 | 1.273 | 1.869 |

**Confidence $\delta$ and number of samples:** All our experiments are performed with $\delta = 0.001$. In NAIVE and CLUSTER, we follow our theoretical results and sample $N = \frac{1}{\delta} \log \frac{1}{\delta^2}$ points from each group. Choosing the training samples for the baselines is a bit more tricky. For these regression tasks a training point is a tuple $(x, y, \sigma(x, y))$. In other words, these tasks do not distinguish between queries and samples. To be as fair as possible when comparing against our algorithms, we provide the baselines with $N + Q$ training points of the form $(x, y, \sigma(x, y))$, where $Q$ is the maximum number of queries used in any of our algorithms.

**Testing:** We test our algorithms over $1,000$ trials, where each trial consists of independently sampling two elements $x, y$, one for each group, and then inputting those to the predictors. We are interested in two metrics. The first is the relative error percentage; if $p$ is the prediction for elements $x, y$ and $t$ is their true similarity value, the relative error percentage is $100 \cdot |p - t|/t$. The second metric we consider is the absolute error divided by $\epsilon$; if $p$ is the prediction for two elements and $t$ is their true similarity value, this metric is $|p - t|/\epsilon$. We are interested in the latter metric because our theoretical guarantees are of the form $|f(x, y) - \sigma(x, y)| = O(\epsilon)$.

Table 1 shows the average relative and absolute value error, together with the standard deviation for these metrics across all $1,000$ runs. It is clear that our algorithms dominate the baselines since they exhibit smaller errors with smaller standard deviations. In addition, NAIVE appears to have a tiny edge over CLUSTER, and this is expected (the queries of CLUSTER are a subset of NAIVE's). However, as shown in Table 2, CLUSTER leads to a significant decrease in oracle queries compared to NAIVE, thus justifying its superiority. To further demonstrate the statistical behavior of the errors, we present Figure 2. This depicts the empirical CDF of each error metric through a bar plot. Specifically, the height of each bar corresponds to the fraction of test instances whose error is at most the value in the x-axis directly underneath the bar. Once again, we see that our algorithms outperform the baselines, since their corresponding bars are always higher than those of the baselines.

## Disclaimer

Table 2: Percent of decrease in queries when using CLUSTER instead of NAIVE

| Credit Card Default | Adult | Give Me Some Credit |
|---|---|---|
| 81.01% | 80.40% | 84.87% |

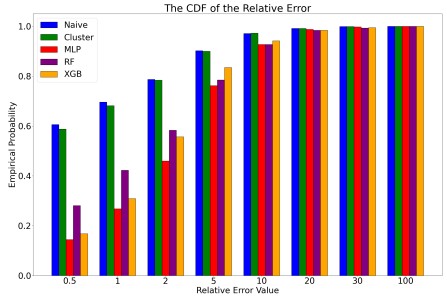
(a) Relative Error Percentage

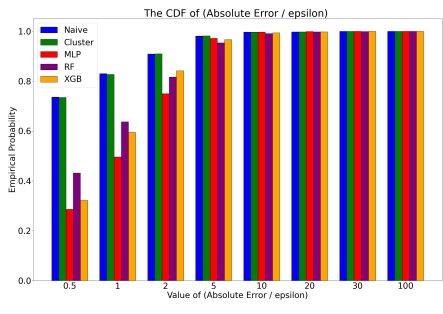
(b) (Absolute Error) / $\epsilon$

Figure 2: Empirical CDF for Credit Card Default

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

# A Missing proofs

***Proof of Theorem*** *4.2*. For two distinct groups $\ell$ and $\ell'$, consider what will happen when we are asked to compare some $x \in \mathcal{D}_\ell$ and $y \in \mathcal{D}_{\ell'}$. Properties $\mathcal{M}_1$ and $\mathcal{M}_2$ of $\sigma_{\ell,\ell'}$ imply

$$Q \le \sigma_{\ell,\ell'}(x,y) \le P, \text{ where}$$
$$P := d_\ell(x, \pi(x)) + \sigma_{\ell,\ell'}(\pi(x), \pi(y)) + d_{\ell'}(y, \pi(y))$$
$$Q := \sigma_{\ell,\ell'}(\pi(x), \pi(y)) - d_\ell(x, \pi(x)) - d_{\ell'}(y, \pi(y))$$

Note that when $d_\ell(x, \pi(x)) \le 3\epsilon$ and $d_{\ell'}(y, \pi(y)) \le 3\epsilon$, the above inequalities and the definition of $f_{\ell,\ell'}(x,y)$ yield $\left| f_{\ell,\ell'}(x,y) - \sigma_{\ell,\ell'}(x,y) \right| \le 6\epsilon$. Thus, we just need upper bounds for $\mathcal{A} := \Pr_{S_\ell, x \sim \mathcal{D}_\ell}[\forall x' \in S_\ell : d(x, x') > 3\epsilon]$ and $\mathcal{B} := \Pr_{S_{\ell'}, y \sim \mathcal{D}_{\ell'}}[\forall y' \in S_\ell : d(y, y') > 3\epsilon]$, since the previous analysis and a union bound give $\Pr[\text{Error}_{(\ell,\ell')}] \le \mathcal{A} + \mathcal{B}$. In what follows we present an upper bound for $\mathcal{A}$. The same analysis gives an identical bound for $\mathcal{B}$.

Before we proceed to the rest of the proof, we have to provide an existential construction. For the sake of simplicity *we will be using the term dense for elements of $\mathcal{D}_\ell$ that are not $(\epsilon, \delta)$-rare*. For every $x \in \mathcal{D}_\ell$ that is dense, we define $B_x := \{x' \in \mathcal{D}_\ell : d_\ell(x, x') \le \epsilon\}$. Observe that the definition of dense elements implies $\Pr_{x' \sim \mathcal{D}_\ell}[x' \in B_x] \ge \delta$ for every dense $x$. Next, consider the following process. We start with an empty set $\mathcal{R} = \{\}$, and we assume that all dense elements are unmarked. Then, we choose an arbitrary unmarked dense element $x$, and we place it in the set $\mathcal{R}$. Further, for every dense $x' \in \mathcal{D}_\ell$ that is unmarked and has $B_x \cap B_{x'} \ne \emptyset$, we mark $x'$ and set $\psi(x') = x$. Here the function $\psi$ maps dense elements to elements of $\mathcal{R}$. We continue this picking process until all dense elements have been marked. Since $B_z \cap B_{z'} = \emptyset$ for any two $z, z' \in \mathcal{R}$ and $\Pr_{x' \sim \mathcal{D}_\ell}[x' \in B_z] \ge \delta$ for $z \in \mathcal{R}$, we have $|\mathcal{R}| \le 1/\delta$. Also, for every dense $x$ we have $d_\ell(x, \psi(x)) \le 2\epsilon$ due to $B_x \cap B_{\psi(x)} \ne \emptyset$.

Now we are ready to upper bound $\mathcal{A}$.

$$\mathcal{C} := \Pr_{S_\ell, x \sim \mathcal{D}_\ell}[\forall x' \in S_\ell : d(x, x') > 3\epsilon \wedge x \text{ is } (\epsilon, \delta)\text{-rare}]$$
$$\le \Pr_{x \sim \mathcal{D}_\ell}[x \text{ is } (\epsilon, \delta)\text{-rare}] = p_\ell(\epsilon, \delta)$$
$$\mathcal{D} := \Pr_{S_\ell, x \sim \mathcal{D}_\ell}[\forall x' \in S_\ell : d(x, x') > 3\epsilon \wedge x \text{ is dense}]$$
$$\le \Pr_{S_\ell}[\exists r \in \mathcal{R} : B_r \cap S_\ell = \emptyset]$$
$$\le \sum_{r \in \mathcal{R}} \Pr_{S_\ell}[B_r \cap S_\ell = \emptyset] \le |\mathcal{R}|(1-\delta)^{|S_\ell|} \le |\mathcal{R}|e^{-\delta|S_\ell|} \le \delta$$

The upper bound for $\mathcal{C}$ is trivial. We next explain the computations for $\mathcal{D}$. For the transition between the first and the second line we use a proof by contradiction. Hence, suppose that $S_\ell \cap B_r \ne \emptyset$ for every $r \in \mathcal{R}$, and let $i_r$ denote an arbitrary element of $S_\ell \cap B_r$. Then, for any dense element $x \in \mathcal{D}_\ell$ we have $d_\ell(x, \pi(x)) \le d_\ell(x, i_{\psi(x)}) \le d_\ell(x, \psi(x)) + d_\ell(\psi(x), i_{\psi(x)}) \le 2\epsilon + \epsilon = 3\epsilon$. Back to the computations for $\mathcal{D}$, to get the third line we simply used a union bound. To get from the third to the fourth line, we used the definition of $r \in \mathcal{R}$ as a dense element, which implies that the probability of sampling any element of $B_r$ in one try is at least $\delta$. The final bound is a result of numerical calculations using $|\mathcal{R}| \le \frac{1}{\delta}$ and $|S_\ell| = \frac{1}{\delta} \log \frac{1}{\delta^2}$.

To conclude the proof, observe that $\mathcal{A} = \mathcal{C} + \mathcal{D}$, and using a similar reasoning as the one in upper-bounding $\mathcal{A}$ we also get $\mathcal{B} \le p_{\ell'}(\epsilon, \delta) + \delta$. $\qquad\square$

***Proof of Theorem*** *2.3*. Given any $\epsilon, \delta$, consider the following instance of the problem. We have two groups represented by the distributions $\mathcal{D}_1$ and $\mathcal{D}_2$. For the first group we have only one element belonging to it, and let that element be $x$. In other words, every time we draw an element from $\mathcal{D}_1$ that element turns out to be $x$, i.e., $\Pr_{x' \sim \mathcal{D}_1}[x' = x] = 1$. For the second group we have that every $y \in \mathcal{D}_2$ appears with probability $\frac{1}{|\mathcal{D}_2|}$, and $|\mathcal{D}_2|$ is a huge constant $c \gg 0$, with $\frac{1}{c} \ll \delta$.

Now we define all similarity values. At first, the similarity function for $\mathcal{D}_1$ will trivially be $d_1(x, x) = 0$. For the second group, for every distinct $y, y' \in \mathcal{D}_2$ we define $d_2(y, y') = 1$. Obviously, for every $y \in \mathcal{D}_2$ we set $d_2(y, y) = 0$. Observe that $d_1$ and $d_2$ are metric functions for their respective groups. As for the across-groups similarities, each $\sigma(x, y)$ for $y \in \mathcal{D}_2$ is chosen independently, and it is

drawn uniformly at random from $[0, 1]$. Note that this choice of $\sigma$ satisfies the necessary metric-like properties $\mathcal{M}_1$ and $\mathcal{M}_2$ that were introduced in Section 2.

Further, since $\epsilon, \delta \in (0, 1)$, any $y \in \mathcal{D}_2$ will be $(\epsilon, \delta)$-rare:

$$\Pr_{y' \sim \mathcal{D}_2}[d_2(y, y') \le \epsilon] = \Pr_{y' \sim \mathcal{D}_2}[y' = y] = \frac{1}{|\mathcal{D}_2|} < \delta$$

The first equality is because the only element within distance $\epsilon$ from $y$ is $y$ itself. The last inequality is because $\frac{1}{|\mathcal{D}_2|} < \delta$. Therefore, since all elements are $(\epsilon, \delta)$-rare, we have $p_2(\epsilon, \delta) = 1$.

Consider now any learning algorithm that produces an estimate function $f$. For any $y \in \mathcal{D}_2$, let us try to analyze the probability of having $|f(x, y) - \sigma(x, y)| = \omega(\epsilon)$. At first, note that when $y \in S_2$, we can always get the exact value $f(x, y)$, since $x$ will always be in $S_1$. The probability of having $y \in S_2$ is $1 - (1 - 1/|\mathcal{D}_2|)^N$, where $N$ is the number of used samples. Since we have control over $|\mathcal{D}_2|$ when constructing this instance, we can always set it to a large enough value that will give $1 - (1 - 1/|\mathcal{D}_2|)^N = \epsilon$; note that this is possible because $1 - (1 - 1/|\mathcal{D}_2|)^N$ is decreasing in $|\mathcal{D}_2|$ and $\lim_{|\mathcal{D}_2| \to \infty} \left(1 - (1 - 1/|\mathcal{D}_2|)^N\right) = 0$. Hence,

$$\Pr_y[|f(x, y) - \sigma(x, y)| = \omega(\epsilon)] = (1 - \epsilon) \Pr_y[|f(x, y) - \sigma(x, y)| = \omega(\epsilon) \mid y \notin S_2] + \epsilon \quad (3)$$

When $y$ will not be among the samples, the algorithm needs to learn $\sigma(x, y)$ via some other value $\sigma(x, y')$, for $y'$ being a sampled element of the second group. However, due to the construction of $\sigma$ the values $\sigma(x, y)$ and $\sigma(x, y')$ are independent. This means that knowledge of any $\sigma(x, y')$ (with $y \ne y'$) provides no information at all on $\sigma(x, y)$. Thus, the best any algorithm can do is guess $f(x, y)$ uniformly at random from $[0, 1]$. This yields $\Pr[|f(x, y) - \sigma(x, y)| = \omega(\epsilon) \mid y \notin S_2] = 1 - \Pr[|f(x, y) - \sigma(x, y)| = O(\epsilon) \mid y \notin S_2] = 1 - O(\epsilon) = p_2(\epsilon, \delta) - O(\epsilon)$. Combining this with (3) gives the desired result. □

***Proof of Theorem*** *4.3.* For two distinct groups $\ell$ and $\ell'$, consider comparing some $x \in \mathcal{D}_\ell$ and $y \in \mathcal{D}_{\ell'}$. To begin with, let us assume that $d_\ell(x, \pi(x)) \le 3\epsilon$ and $d_{\ell'}(y, \pi(y)) \le 3\epsilon$. Furthermore, the execution of the algorithm implies $H^\ell_{\pi(x)} \cap H^\ell_{r_\ell(\pi(x))} \ne \emptyset$, and thus the triangle inequality and the definitions of the sets $H^\ell_{\pi(x)}, H^\ell_{r_\ell(\pi(x))}$ give $d_\ell(\pi(x), r_\ell(\pi(x))) \le 16\epsilon$. Similarly $d_{\ell'}(\pi(y), r_{\ell'}(\pi(y))) \le 16\epsilon$. Eventually:

$$d_\ell(x, r_\ell(\pi(x))) \le d_\ell(x, \pi(x)) + d_\ell(\pi(x), r_\ell(\pi(x))) \le 19\epsilon$$
$$d_{\ell'}(y, r_{\ell'}(\pi(y))) \le d_{\ell'}(y, \pi(y)) + d_{\ell'}(\pi(y), r_{\ell'}(\pi(y))) \le 19\epsilon$$

For notational convenience, let $A := d_\ell(x, r_\ell(\pi(x)))$ and $B := d_{\ell'}(y, r_{\ell'}(\pi(y)))$. Then, the metric properties $\mathcal{M}_1$ and $\mathcal{M}_2$ of $\sigma_{\ell,\ell'}$ and the definition of $f_{\ell,\ell'}(x, y)$ yield

$$|\sigma_{\ell,\ell'}(x, y) - f_{\ell,\ell'}(x, y)| \le A + B \le 38\epsilon$$

Overall, we proved that when $d_\ell(x, \pi(x)) \le 3\epsilon$ and $d_{\ell'}(y, \pi(y)) \le 3\epsilon$, we have $\left|f_{\ell,\ell'}(x, y) - \sigma_{\ell,\ell'}(x, y)\right| \le 38\epsilon$. Finally, as shown in the proof of Theorem 4.2, the probability of not having $d_\ell(x, \pi(x)) \le 3\epsilon$ and $d_{\ell'}(y, \pi(y)) \le 3\epsilon$, is at most $2\delta + p_\ell(\epsilon, \delta) + p_{\ell'}(\epsilon, \delta)$. □

***Proof of Lemma*** *4.5.* Consider a group $\ell$, and let $C^*$ be its optimal solution for the problem of Definition 4.4. We first claim that each $\mathcal{H}^\ell_c$ with $c \in C^*$ contains at most one element of $R_\ell$. This is due to the following. For any $c \in C^*$, we have $d_\ell(z, z') \le d_\ell(z, c) + d_\ell(c, z') \le 8\epsilon$ for all $z, z' \in \mathcal{H}^\ell_c$. In addition, the construction of $R_\ell$ trivially implies $d_\ell(x, x') > 8\epsilon$ for all $x, x' \in R_\ell$. Thus, no two elements of $R_\ell$ can be in the same $\mathcal{H}^\ell_c$ with $c \in C^*$. Finally, since Definition 4.4 requires all $x \in R_\ell$ to be covered, we have $|R_\ell| \le |C^*| = OPT_\ell$. □

***Proof of Lemma*** *4.6.* Consider a group $\ell$. Initially, through Definition 4.4 it is clear that $OPT_\ell \le |S_\ell| = N$. For the second statement of the lemma we need to analyze $OPT_\ell$ in a more clever way.

Recall the classification of elements $x \in \mathcal{D}_\ell$ that was first introduced in the proof of Theorem 4.2. According to this, an element can either be $(\epsilon, \delta)$-rare, or dense. Now we will construct a solution $C_\ell$ to the problem of Definition 4.4 as follows.

At first, let $S_{\ell,r}$ be the set of $(\epsilon, \delta)$-rare elements of $S_\ell$. We will include all of $S_{\ell,r}$ to $C_\ell$, so that all $(\epsilon, \delta)$-rare elements of $S_\ell$ are covered by $C_\ell$. Further:

$$\mathbb{E}\big[|S_{\ell,r}|\big] = p_\ell(\epsilon, \delta)N \tag{4}$$

Moving on, recall the construction shown in the proof of Theorem 4.2. According to that, there exists a set $\mathcal{R}$ of at most $\frac{1}{\delta}$ dense elements from $\mathcal{D}_\ell$, and a function $\psi$ that maps every dense element $x \in \mathcal{D}_\ell$ to an element $\psi(x) \in \mathcal{R}$, such that $d_\ell(x, \psi(x)) \leq 2\epsilon$. Let us now define for each $x \in \mathcal{R}$ a set $G_x := \{x' \in \mathcal{D}_\ell : x' \text{ is dense and } \psi(x') = x\}$, and note that $d_\ell(z, z') \leq d_\ell(z, x) + d_\ell(z', x) \leq 4\epsilon$ for all $z, z' \in G_x$. Thus, for each $x \in \mathcal{R}$ with $G_x \cap S_\ell \neq \emptyset$, we place in $C_\ell$ an arbitrary $y \in G_x \cap S_\ell$, and that $y$ gets all of $G_x \cap S_\ell$ covered. Finally, since the sets $G_x$ induce a partition of the dense elements of $\mathcal{D}_\ell$, $C_\ell$ covers all dense elements of $S_\ell$.

Equation (4) and $|\mathcal{R}| \leq \frac{1}{\delta}$ yield $\mathbb{E}\big[|C_\ell|\big] \leq \frac{1}{\delta} + p_\ell(\epsilon, \delta)N$. Also, since $C_\ell$ is shown to be a feasible solution for problem of Definition 4.4, we get

$$OPT_\ell \leq |C_\ell| \implies \mathbb{E}[OPT_\ell] \leq \frac{1}{\delta} + p_\ell(\epsilon, \delta)N \qquad \square$$

***Proof of Theorem 2.5***. We are given accuracy and confidence parameters $\epsilon, \delta \in (0, 1)$ respectively. For the sake of simplifying the exposition in the proof, let us assume that $\frac{1}{\delta}$ is an integer; all later arguments can be generalized in order to handle the case of $1/\delta \notin \mathbb{N}$.

We construct the following problem instance. We have two groups represented by the distributions $\mathcal{D}_1$ and $\mathcal{D}_2$. In addition, for both of these groups we assume that the support of the corresponding distribution contains $\frac{1}{\delta}$ elements, and $\Pr_{x' \sim \mathcal{D}_1}[x = x'] = \delta$ for every $x \in \mathcal{D}_1$ as well as $\Pr_{y' \sim \mathcal{D}_2}[y = y'] = \delta$ for every $y \in \mathcal{D}_2$.

For every $x, x' \in \mathcal{D}_1$ let $d_1(x, x') = 1$, and $d_1(x, x) = 0$ for every $x \in \mathcal{D}_1$. Similarly, for every $y, y' \in \mathcal{D}_2$ we set $d_2(y, y') = 1$, and for every $y \in \mathcal{D}_2$ we set $d_2(y, y) = 0$. The functions $d_1$ and $d_2$ are clearly metrics. As for the across-groups similarity values, each $\sigma(x, y)$ for $x \in \mathcal{D}_1$ and $y \in \mathcal{D}_2$ is chosen independently, and it is drawn uniformly at random from $[0, 1]$. Note that this choice of $\sigma$ satisfies the necessary properties $\mathcal{M}_1, \mathcal{M}_2$ introduced in Section 2.

In this proof we are also focusing on a more special learning model. In particular, we assume that the distributions $\mathcal{D}_1$ and $\mathcal{D}_2$ are known. Hence, there is no need for sampling. The only randomness here is over the random arrivals $x \sim \mathcal{D}_1$ and $y \sim \mathcal{D}_2$, where $x$ and $y$ are the elements that need to be compared. Obviously, the similarity function $\sigma$ would still remain unknown to any learner. Finally, the queries required for learning in this model cannot be more than the queries required in the original model, and this is because this model is a special case of the original.

Consider now an algorithm with the error guarantees mentioned in the Theorem statement, and focus on a fixed pair $(x, y)$ with $x \in \mathcal{D}_1$ and $y \in \mathcal{D}_2$. If the algorithm has queried the oracle for $(x, y)$, it knows $\sigma(x, y)$ with absolute certainty. Let us study what happens when the algorithm has not queried the oracle for $(x, y)$. In this case, because the values $\sigma(x', y')$ with $x' \in \mathcal{D}_1$ and $y' \in \mathcal{D}_2$ are independent, no query the algorithm has performed can provide any information for $\sigma(x, y)$. Thus, the best the algorithm can do is uniformly at random guess a value in $[0, 1]$, and return that as the estimate for $\sigma(x, y)$. If $\bar{\mathcal{Q}}_{x,y}$ denotes the event where no query is performed for $(x, y)$, then

$$\begin{aligned}
\mathcal{P} &:= \Pr\big[|f_{\ell,\ell'}(x, y) - \sigma_{\ell,\ell'}(x, y)| = \omega(\epsilon) \mid \bar{\mathcal{Q}}_{x,y}\big] \\
&= 1 - \Pr\big[|f_{\ell,\ell'}(x, y) - \sigma_{\ell,\ell'}(x, y)| = O(\epsilon) \mid \bar{\mathcal{Q}}_{x,y}\big] \\
&= 1 - O(\epsilon) \\
&= \Omega(1)
\end{aligned}$$

where the randomness comes only from the algorithm.

For the sake of contradiction, suppose the algorithm uses $q = o(1/\delta^2)$ queries. Since each $(x, y)$ is equally likely to appear for a comparison, the overall error probability is

$$\left(\frac{1/\delta^2 - q}{1/\delta^2}\right) \cdot \mathcal{P} = \Omega(1) \cdot \Omega(1) = \Omega(1)$$

Contradiction; the error probability was assumed to be $O(\delta)$. $\qquad \square$

***Proof of Corollary*** *4.8.* When every $p_\ell(\epsilon, \delta)$ is very close to 0, Lemma 4.6 gives $\mathbb{E}[OPT_\ell] \leq \frac{1}{\delta}$. Thus, by inequality (1) the expected queries are $\frac{\gamma(\gamma-1)}{\delta^2}$. Theorem 2.5 concludes the proof. $\qquad\square$

**Theorem A.1.** *The function $\sigma$ defined in Equation (2) is not metric, and satisfies properties $\mathcal{M}_1, \mathcal{M}_2$ for $d_1(x, y) = d_2(x, y) = \sqrt{\sum_{i \in [d]} (x_i - y_i)^2}$, when $\alpha_i, \beta_i \in [0, 1]$ for all $i \in [d]$.*

*Proof.* The easiest way to see that $\sigma(x, y)$ is not a metric, is by realizing that it does not satisfy the symmetry property and also $x = y$ does not necessarily imply $\sigma(x, y) = 0$. Both of these issues stem from the offsets $\theta_i$. To verify that symmetry is violated let $x, y$ be 1-dimensional, and let $x = 1, y = 2, \alpha = 1, \beta = 1, \theta = 2$. Then $\sigma(1, 2) = 1$, while $\sigma(2, 1) = 3$. Next, let $x = 1, y = 1, \alpha = 1, \beta = 1, \theta = 1$. In this example, although $x = y$ we have $\sigma(x, y) = 1 \neq 0$.

In the following we are going to show that $\sigma$ satisfies $\mathcal{M}_1$. The proof for $\mathcal{M}_1$ is identical. At first, take $x, y, z \in \mathbb{R}^d$, such that $x, z \in \mathcal{D}_1$ and $y \in \mathcal{D}_2$. Then:

$$\sigma(x, y) = \sqrt[3]{\sum_{i \in [d]} |\alpha_i \cdot x_i - \beta_i \cdot y_i + \theta_i|^3} = \sqrt[3]{\sum_{i \in [d]} |(\alpha_i \cdot x_i - \alpha_i \cdot z_i) + (\alpha_i \cdot z_i - \beta_i \cdot y_i + \theta_i)|^3}$$

$$\leq \sqrt[3]{\sum_{i \in [d]} |(\alpha_i \cdot x_i - \alpha_i \cdot z_i)|^3} + \sqrt[3]{\sum_{i \in [d]} |(\alpha_i \cdot z_i - \beta_i \cdot y_i + \theta_i)|^3}$$

$$= \sqrt[3]{\sum_{i \in [d]} a_i^3 |x_i - z_i|^3} + \sigma(z, y)$$

$$\leq \sqrt[3]{\sum_{i \in [d]} |x_i - z_i|^3} + \sigma(z, y)$$

$$\leq d_1(x, z) + \sigma(z, y)$$

To get the first inequality we used the triangle inequality for $\ell_3$. The second to last inequality is because $\alpha_i^3 \leq 1$ for all $i$, and the last inequality is because the $\ell_3$ norm of a vector is always smaller than its $\ell_2$ norm. $\qquad\square$

# B   Additional experimental results

Table 3: Error statistics for Adult

| Algorithm | Average Relative Error % | SD of Relative Error % | Average (Absolute Error)/$\epsilon$ | SD of (Absolute Error)/$\epsilon$ |
|---|---|---|---|---|
| NAIVE | 1.319 | 3.381 | 0.847 | 2.382 |
| CLUSTER | 1.321 | 3.383 | 0.849 | 2.383 |
| MLP | 4.119 | 6.092 | 2.306 | 1.977 |
| RF | 3.519 | 6.657 | 2.020 | 2.987 |
| XGB | 2.248 | 5.351 | 1.215 | 1.655 |

Table 4: Error statistics for Give Me Some Credit

| Algorithm | Average Relative Error % | SD of Relative Error % | Average (Absolute Error)/$\epsilon$ | SD of (Absolute Error)/$\epsilon$ |
|---|---|---|---|---|
| NAIVE | 1.630 | 4.106 | 2.644 | 7.710 |
| CLUSTER | 1.645 | 4.118 | 2.648 | 7.665 |
| MLP | 5.819 | 9.073 | 7.588 | 8.750 |
| RF | 5.692 | 12.404 | 7.258 | 34.508 |
| XGB | 5.614 | 13.355 | 6.516 | 7.754 |

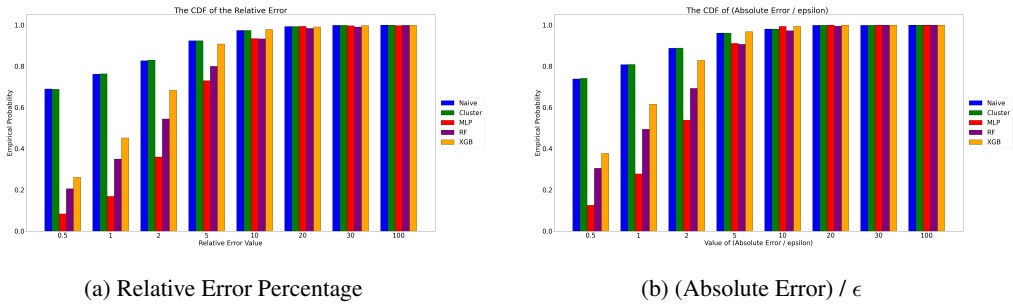

(a) Relative Error Percentage        (b) (Absolute Error) / $\epsilon$

Figure 3: Empirical CDF for Adult

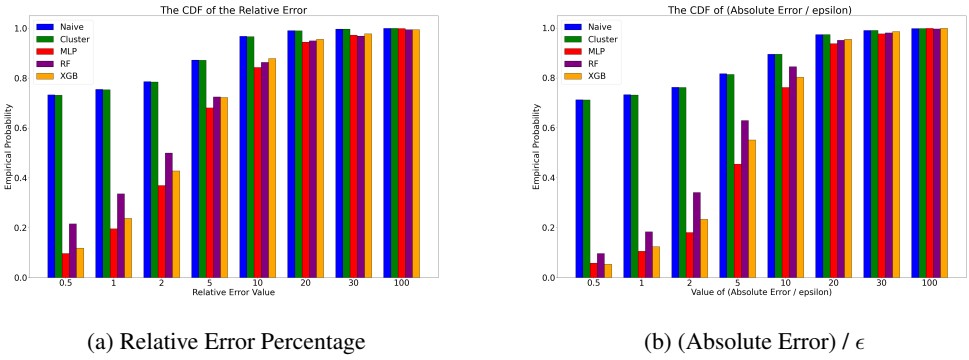

(a) Relative Error Percentage        (b) (Absolute Error) / $\epsilon$

Figure 4: Empirical CDF for Give Me Some Credit

