# OpenReview forum: "Comparing Apples to Oranges: Learning Similarity Functions for Data Produced by Different Distributions"
_NeurIPS.cc/2023/Conference — NeurIPS 2023 poster_

### Official Review · Reviewer_L6oz · 2023-06-18

**Soundness:** 2 fair
**Presentation:** 3 good
**Contribution:** 2 fair
**Rating:** 4
**Confidence:** 3

**Summary:**

This work considers PAC-learnable algorithms for estimating a intra-group
similarity function where:
1. intra-group similarity functions are metrics and given in advance,
2. inter-group similarities satisfy a notion of the triangle inequality.
It is assumed that points are always labeled by the group to which they belong.

The authors present two algorithms Naive and Cluster, and
give PAC-learning guarantees for them (i.e., the sample complexity needed for
the learned similarity function to be $\varepsilon$-close to the true
similarity function with probability at least $\delta$). The clustering
algorithm uses a ``representative'' subset of each groups samples before
querying inter-group similarities to reduce the query complexity. The authors
give substantive experiments to empirically study their algorithms.

**Strengths:**

- The "simple learning algorithm" is indeed simple and clean with strong
  guarantees (Theorem 2.1).
- The idea of using $k$ centers from each group's samples as representatives is
  a natural improvement over the simple algorithm.
- Very good experiments where the authors define a challenging similarity
  function to try to recover:
  * Creating two groups based on the "married" feature is an excellent instance
    of what the paper aims to study.
  * The benchmark algorithms MLP, RF, and XGB each see at least as many samples
    as the proposed algorithms, and are therefore fair (assuming they fully
    trained).

**Weaknesses:**

- In section 2.1, you say you begin with a simple algorithm, but it is never
  described before Theorem 2.1. Consider giving a quick description to the
  reader before they take in the theorem statements.
- The presentation could be improved slightly: less motivation in the intro and
  more details in the main text (e.g., proof sketches).

**Questions:**

- [line 22] If two objects $x$ and $y$ are identical, they have large
  similarity? Maybe it makes sense to use the distance-version of $\sigma$ here
  so that it is clear the upper bound on the difference is zero for identical
  objects.
- [line 171] Is $Q_\ell$ a random variable? If so, it might be worthwhile to
  make that more explicit.
- [line 229] Is a factor of $1/2$ missing in the query complexity? Are we
  assuming that $\sigma(x,y)$ is symmetric?
- Does the similarity function in Eq. (2) have the property that inter-group
  points are further away on average than intra-group points?

---

> ### Author Rebuttal · Authors · 2023-08-04
>
> Thank you for your review. We will certainly take your feedback into consideration.
>
> We first address the weaknesses that you mentioned:
>
>    1. Even though there is a description of the simple algorithm in lines 225-228, we will try to make it more explicit.
>    2. We will try to work on the overall presentation as you suggested, giving more details in the main text.
>
> Addressing the questions:
>
>    1. If two objects are identical, then their similarity/distance is 0. This basically shows that the two objects are in the same position in the space.
>    2. $Q_\ell$ is indeed a random variable. We can explicitly mention this in Theorem 2.4.
>    3. You are right. This is a typo that we can fix.
>    4. Our math works even when $\sigma(x,y)$ is not symmetric. However, having it being symmetric is a natural assumption in real application. Nonetheless, our math can work for a more general case (not necessarily symmetric).
>    5. "Does the similarity function in Eq. (2) have the property that inter-group points are further away on average than intra-group points?" Not necessarily. We do not need to make such an assumption. Our math works regardless of that.
>
> To conclude, we would appreciate if the reviewer reconsiders their score. **We feel like the weaknesses that are mentioned are not substantial enough to justify a reject rating.**

---

> > ### Comment · Reviewer_L6oz · 2023-08-13
> >
> > I have read all the reviews and author responses, and will keep my rating the same.

---

### Official Review · Reviewer_voHg · 2023-07-05

**Soundness:** 3 good
**Presentation:** 3 good
**Contribution:** 3 good
**Rating:** 7
**Confidence:** 3

**Summary:**

The paper is about a mechanism to compare elements  of a set that belong to two distinct distributions when two distinct sets of features for each distribution.   This is a natural problem that arises in many practical settings.  The paper presents the underlying problem and the formal framework with some guarantees assuming some 3rd database (oracle) has answers about comparability of any pair of elements.

**Strengths:**


Compared to other papers I have to review for NeurIPS this year, the authors of this paper go out of their way to explain what they're trying to do and motivate the theoretical results they obtain.

The topic is a good one -- often the same data appear in different databases with different attributes and this paper provides a mechanism to construct a similarity function across different features *assuming* there is an oracle that can compare across different features.



**Weaknesses:**



The basic idea of this paper is well explained and illustrated.  However, unlike much of the rest of the discussion in this paper, the "no free lunch" theorem is not explored in full.  A key question anyone would have is how rare or common is it for max { p_l, p_l' } is large and therefore the framework introduced in this paper is not applicable?  The authors could use the datasets they used for experimental results to shed some light on this question.  This is actually the 1st question anyone wishing to use this result would be asking themselves.

**Questions:**


See above.

**Limitations:**


The major limitation of this work is the assumption of the availability of an oracle that can provide the ground truth similarity between any pair of elements of the set across a broader set of features than what's given.  Eg, in the sport (say) of 100 meter sprint, on the one hand we have the age, height, weight, sex/gender, ethnicity, etc., of the competitors and on the other hand, we have the diet and medications for some.   Naturally a good similarity function to use to predict the athletes' performance needs to incorporate both sets of features.  So in such cases, where does one look for this all-knowing oracle even if the number of queries to it are limited?

---

> ### Author Rebuttal · Authors · 2023-08-04
>
> Thank you so much for your positive comments. We are glad to see our work getting appreciated.
>
> Regarding the weakness you mentioned. This is indeed a great question and we can certainly investigated it even more. Our experimental results show that $\max (p_\ell(\epsilon, \delta), p_{\ell'}(\epsilon, \delta))$ is indeed small in the datasets we tested (otherwise we wouldn't be achieving accurate results), however we did not explicitly quantify how small this value is. Nonetheless, this is actually easy to do. Since we have the bound of the algorithm and we know $\delta$, we can work backwards and compute this quantity. Specifically:
>
>    1. In Credit Card Default and Adult this value is around 0.01
>    2. In Give me Some Credit this value is approximately 0.03

---

> > ### Comment · Reviewer_voHg · 2023-08-18
> >
> > Thank you for your clarification. I'll keep my rating as is.

---

### Official Review · Reviewer_3J4J · 2023-07-07

**Soundness:** 4 excellent
**Presentation:** 4 excellent
**Contribution:** 4 excellent
**Rating:** 8
**Confidence:** 4

**Summary:**

This paper studies the problem of learning a similarity function of items generated from different distributions. Precisely, there are two groups of items, and the items in each group follow a specific distribution and the group has an intra-cluster metric. This paper wants to learn a cross-group metric from limited number of samples with fairness and PAC error guarantee. This problem is critical for creating fair comparisons of items in heterogeneous groups.
Further, this paper provides upper bound and lower bound analysis on both i) error rates of algorithms with polynomial sample complexity; and ii) the sample complexity of a PAC (\epsilon,\delta) algorithm. From the analysis, I think the bounds on both error rates and sample complexity are reasonably good though still have rooms for improvements.
Finally, the authors present the numerical results on real-world data sets. The numerical results show out-performed performance of the proposed algorithms.


**Strengths:**

Novelty: This paper studies an interesting, novel, and theoretical-fundamental problem. The algorithms are not very new, but already show good enough performance in this new problem, so it is okay. Overall, this paper is of high novelty.

Quality: The paper has a high quality. The results are complete. It is not common to see well-established lower bound analysis, but this paper does it for both error rates and sample complexity. Also, the upper bounds and lower bounds are reasonably close, and the gaps between them are not very high, which is good enough for initial research. The paper also conducts nice numerical experiments on real-world data to confirm the theoretical conclusions, which strengthen this paper's quality.

Clarity: This paper is well written. It has a clear story-telling: it introduces the problems, explain the mathematical formulation, state the theoretical results, and present the proofs well.

Significance: In my personal opinion, this is a very important problem in terms of theory, as it is actually understanding the fundamental learnability of aggregating two or multiple metric spaces together. In terms of practice, I am not very confident, but the examples such as the student case given in the paper are convincing. Overall, the I think this paper is significant enough.

**Weaknesses:**

No major weakness found. The paper is good overall. One minor question: How often would the real-world scenarios have non-complete metrics, as this is an improvement of this paper?

**Questions:**

As stated in Weaknesses.

---

> ### Author Rebuttal · Authors · 2023-08-04
>
> We sincerely thank the reviewer for their very positive feedback. We are glad to see that our work gets appreciated.
>
> With regards to your question, we hypothesize that in practice, the vast majority of application the similarity scores will not be forming a metric. Take for instance our second motivating example, where we have two distinct data sources. In this case, experts usually have a "private" transformation function that embeds one feature space to the other. This transformation will most likely be based on domain knowledge taking the form of complex rules, e.g., if features $S$ (a being a set of features) in the first space take a values $f(S) \in \mathbb{R}^{|S|}$, then features $W$ in the second space should take values $g(f(S)) \in \mathbb{R}^{|W|}$ ($W$ being a set of features). Clearly such complex transformation will most likely result in arbitrary non-metric space.
>
> For fairness related applications, we refer to a comment in the seminal paper of Dwork et al., where it is mentioned that such similarity functions are not necessarily metric.

---

### Official Review · Reviewer_qRef · 2023-07-25

**Soundness:** 1 poor
**Presentation:** 3 good
**Contribution:** 1 poor
**Rating:** 2
**Confidence:** 4

**Summary:**

This paper involves finding an inter-group similarity function with (limited) oracle advice when one has access to the intra-group metrics. The authors give theoretical guarantees about their algorithm and claim to have proven lower bounds on the performance of any algorithm.

**Strengths:**

I like this problem. The proposed problem is a nice contribution - but only if its solution improves on the more general learning of the whole metric (without knowledge of the intra-cluster similarities) in the case that the similarities are a metric (the fact that the assumptions of this paper are slightly less restricted than a metric is by itself not enough novelty in my opinion)

The experiments show this algorithm outperforms certain baselines (although similar works are not considered in the experiments).

The paper is well written from a language point of view.

**Weaknesses:**

I do not think that the algorithms themselves are innovative enough for a conference like NeurIPS.

From Line 437 of the appendix it seems that the true definition of a rare element should be one whose probability that ALL of the N samples lie at distance more than epsilon away is greater than delta (not a single sample as is defined in the paper). Although the definition used doesn’t make Theorem 4.1 incorrect it means the bound is much weaker than it could be.

There seems to be something very wrong with Theorem 2.3…
- Firstly, If I limit \delta to 1 then in most cases every element is (\epsilon, \delta) rare so that p_l(\epsilon,delta)=1 and hence the theorem essentially states that with a finite number of samples it is impossible to achieve \epsilon-accuracy with probability no more than \epsilon. This is clearly incorrect.
- Secondly, if the instance space is finite then we can always choose a large enough (but still finite) number of samples such that w.h.p. every possible instance is sampled. If we apply the oracle to every pair we have, w.h.p. (a probability arbitrarily close to 1) found the exact similarity function. This contradicts the theorem.

It is not clear how other works compare to the result of this paper when it is a true metric space (since in this case the multiple distribution idea reduces to a single distribution problem). I do not think that handling the slightly more general non-metric case alone is sufficient for acceptance to a conference like NeurIPS.

In Line 287 the authors state that the metric algorithms cannot be used for non-metric. I suspect that they can still actually run but the bounds don’t necessarily hold (please correct me if I’m wrong). Since there is no theoretical comparison to the metric algorithms (when the similarity is a metric) I would have liked to see empirical comparisons to them in the case that it is a metric.

A minor weakness is that in many applications you would be able to query instances without actually having sampled them.

**Questions:**

Some questions and comments/tips…

Line 81 says that if x is in the support of D_l then it is a member of group l - so can the supports not overlap? This contradicts line 84

In Line 109 I would write “does not need to axiomatically…” - this phrasing points out that this property is a strength rather than a weakness. Maybe also point out that all metric spaces obey your properties.

The functions p_l should be defined before Theorem 2.1 is presented.

In Line 162 it’s not “\epsilon accuracy and a probability at most 1-max{..}” since you use \omega and \Omega.

In Line 74 you mean N^2 not N

**Limitations:**

Authors have addressed limitations.

---

> ### Author Rebuttal · Authors · 2023-08-04
>
> We would like to thank the reviewer for their thoughtful comments.
>
> First we address the mentioned weaknesses of the paper.
>
> 1. Even though are algorithm are relatively simple, their analysis and the analysis of the lower bounds are highly non-trivial.
>
> 2. Here we address the concerns regarding the soundness of Theorems 2.1 and 2.3. Hopefully, after the clarifications it will become clear to the reviewer that  **both theorems are sound and correct**.
>
>    1. The quantities $\mathcal{A}, \mathcal{B}$ in line 437 of the appendix are indeed the ones that you should analyze in order to upper bound the error probability of the algorithm. However, it is not necessary to define $(\epsilon,\delta)$-rarity based on those quantities as mentioned by the reviewer; this will give a very weak bound. Please not that we further analyze $\mathcal{A}, \mathcal{B}$ based on a union bound (lines 451-461), which (for the case of $\mathcal{A}$ since the case of $\mathcal{B}$ is identical) shows that $\mathcal{A} \leq \mathcal{C} + \mathcal{D}$, for two other quantities $\mathcal{C}, \mathcal{D}$. Given the breakdown to $\mathcal{C}, \mathcal{D}$ it is clear that our $(\epsilon,\delta)$-rare definition is **precisely the one needed**. Based on it the upper bound of $\mathcal{C}$ is trivial (simply upper bound by the probability of encountering a rare element), and the bound for $\mathcal{D}$ follows from a union bound over all non-rare elements. **On a high level, what the reviewer is suggesting is a "lazy" definition that would overcome the need for a technical analysis, albeit at the expense of a weaker bound. Our definition is combined with a careful breakdown analysis of the error probability, thus giving a tighter and more meaningful result.**
>
>     2. Thank you for the comment on Theorem 2.3. Although the Theorem itself is indeed correct we believe that the reviewer's comment can help us improve the statement of it and make it more clear. What we show in the proof of this theorem is the following. Given any $\epsilon, \delta \in (0,1)$ and any algorithm that only uses only a finite set of $N$, **we can construct an input instance (distances and distributions) in which the algorithm will necessarily have** $\Pr\big{[} | f_{\ell, \ell'}(x,y) - \sigma_{\ell, \ell'}(x,y)| = \omega(\epsilon) \big{]} = \Omega( \max (p_\ell(\epsilon, \delta), p_{\ell'}(\epsilon, \delta) ) - \epsilon)$. In other words, we show a standard adversarial construction. We have no control over the algorithm and the number of samples it will use, but after seeing those we can give the algorithm a properly constricted instance in which it will fail. **Therefore, if we can always find an failing instance for any algorithm, then the statement of the theorem is true**. As for the counterexamples the reviewer gives:
>
>          1. Note that the first one (all elements are rare) is **exactly** the instance on which we based the construction of the distributions in the proof (see line 475 in the appendix). We precisely show that it is in such an instance that the algorithms would fail (given of course properly constructed distributions).
>          2. When your algorithm uses finitely many samples, the adversary can still construct a distribution of finite support where the algorithm will fail. This is what we do at line 466 in the proof.
>
>           *The problem with both example is that they implicitly assume that the algorithms have knowledge/control of the input instance*. This is not correct however when we are talking about creating an adversarial example (**of course you can solve specific instances, but this does not imply that every instance that is given will be solvable --the adversary gives you exactly those hard instances--**).
>
>    3. We believe that calling the non-metric case slightly more general is a bold and unfair statement. Non-metric functions are **significantly** more complex to work with, and in our opinion this is why they have not been studied in the context of learning similarity for learning (Ilvento [2019], Mukherjee et al. [2020] and Wang et al. [2019] only study metric learning.). The triangle inequality is a powerful tool you can use in algorithmic analysis. As a simple example demonstrating how much more difficult things become in non-metric spaces, take clustering. Non-metric clustering rarely enjoys good theoretical guarantees.
>
>    4. There's nothing prohibiting you from testing a metric learners for this problem. However, 1) it won't yield any theoretical guarantees and 2) in our opinion this is an algorithm designed for a completely different problem. Thus, we chose to use strong regressors as our baselines.
>
> Here we address the questions posed by the reviewer.
>
>    1. In our model an element $x$ can be a member of multiple groups at the same time. There is no contradiction between line 81 and 84. Just because $x$ is let's say a member of the group "Age: 25-40" does not mean that $x$ cannot be a member of the group "High Income". Supports overlap, and every time we have an element $x$ we know which groups it belongs to.
>
>    2. Thank you for the rest of the suggestions. We will make sure we implement them.
>
> To conclude, we hope that our comments clarified why **Theorems 2.1 and 2.3 are indeed correct**, and we wish that the reviewer would increase their score to an accept tier.

---

> > ### Comment · Reviewer_qRef · 2023-08-14
> >
> >
> > Your quantity $p_{\ell}(\epsilon,\delta)$ makes the bound far weaker than if you had left it simply written in terms of $\mathcal{A}$ and $\mathcal{B}$ (which are fine to include in a bound and for which there is no need for further analysis). The fact that your bound is so weak is hidden by Theorem 2.3 which uses a very specifically crafted example where all points are at distance one away from each other and the cardinality is huge (so that $P(x-x’<\epsilon)$ is (relative to the number of samples) tiny and hence the weakness is hidden). More on Theorem 2.3 below.
> >
> > The fact that your bound is extremely weak can be seen by the fact that (in Theorem 4.2) it does not, for any fixed (sufficiently small) $\epsilon$ and any fixed compact space, converge to zero as the number of samples limits to infinity. This is because as $\delta$ limits to zero $p_{\ell}(\epsilon,\delta)$ converges to one. However, in reality, $\mathcal{A}$ and $\mathcal{B}$ converge to zero as the number of samples limits to infinity.
> >
> > The way Theorem 2.3 is (incorrectly) stated in the paper makes the reader totally unaware of what you are free to control in the proof (which is absolutely everything). Although Theorem 2.3 (as you have now described it) may be technically correct it may as well say (in “plain English”) that “any algorithm using a finite number of samples cannot achieve $\epsilon$-accuracy with probability more than (any constant above) zero” - the introduction of $p_{\ell}$ into this statement is completely unnecessary and gives the reader the impression that you have a near-optimal bound.

---

> > > ### Author Response · Authors · 2023-08-14
> > >
> > > *The way Theorem 2.3 is (incorrectly) stated in the paper makes the reader totally unaware of what you are free to control in the proof (which is absolutely everything). Although Theorem 2.3 (as you have now described it) may be technically correct it may as well say (in “plain English”) that “any algorithm using a finite number of samples cannot achieve
> > > -accuracy with probability more than (any constant above) zero” - the introduction of
> > >  into this statement is completely unnecessary and gives the reader the impression that you have a near-optimal bound.*
> > >
> > > Let's say that we are given some computational problem. We consider algorithms $\mathcal{A}$ for it which provide some guarantee $f(\mathcal{A})$, where the smaller the guarantee the better (e.g., in approximation algorithms for minimization problems think of this as the approximation ratio - in our problem $f(\mathcal{A})$ corresponds to the probability of achieving accuracy $\omega(\epsilon)$ [error probability]). Let's say that we want to prove a lower bound for $f(\mathcal{A})$ over all possible algorithms $\mathcal{A}$, for instance that $f(\mathcal{A}) \geq \rho$ (in approximation algorithms this corresponds to the approximability of the problem. For example for k-center you can not have an algorithm with approximation ratio less than 2). One way of doing so, is constructing an instance of the problem, and showing that in this instance the guarantee is necessarily greater than $\rho$, regardless of any algorithm. If we do that then we immediately have $f(\mathcal{A}) \geq \rho$ for any $\mathcal{A}$. How can $f(\mathcal{A}) < \rho$ be true if there's an instance that contradicts it???
> > >
> > > This is precisely the approach we follow in the paper. Which means that the statement of Theorem 2.3 is fine. Namely, for **every** $\epsilon, \delta$ and **every** algorithm that uses some finite sample set, we **construct an instance in which it is impossible to have an error probability lower than $\max(p_\ell, p_{\ell'}) - \epsilon$**. Therefore, if there exists such an instance how can there exist an algorithm that always guarantees error probability of less than $\max(p_\ell, p_{\ell'}) - \epsilon$?
> > >
> > > Now we address the first part of the reviewer's comment.
> > >
> > > The upper bound $\mathcal{A} + \mathcal{B}$ on its own is completely useless. It's completely vague and provides no PAC insight. Our bound has two main benefits:
> > >
> > >    1. A PAC aspect to it through the parameter $\delta$.
> > >    2. The appearance of the parameters $p_\ell$. This captures the essence of what makes the problem hard. These parameters are precisely the input characteristic that affect the feasibility of the problem. Theorem 2.3 clearly shows that. If the $p_\ell$ are large then learning is impossible! Even with a sample size that tends to infinity, there are always instances (where the support of the input distribution is of a larger infinity) where all the elements are isolated and your sampling does not lead to learning.
> > >
> > > Regarding the comment that $\mathcal{A} + \mathcal{B}$ tends to 0 with infinite samples, this is not necessarily true. This is because it might very well be the case that the space is isolated enough so that for every $x$, every other $x'$ is at distance at least $3\epsilon$ from it. For this scenario, $\mathcal{A} + \mathcal{B}$ goes to 1. **The reviewer's suggested tighter analysis here only accounts for an optimistic scenario (the compact case as they said) and not the general case**
> > >
> > > To summarize, the problem is that the reviewer wants to perform the analysis for an optimistic case (everyone has some other point within distance $O(\epsilon)$ from them). This is only a special case and our analysis accounts for the wider picture. By the way, note than in this scenario the analysis of the quantity $\mathcal{C}$ at line 451 can become **a bit** tighter. Nonetheless, this would only work for the special compact case.

---

> > > > ### Author Response · Authors · 2023-08-15
> > > > **Elaborating on our previous comment**
> > > >
> > > > After acknowledging that Theorem 2.3 is correct, the reviewer said that it is incorrectly stated because we do not mention that in the proof we have control of the instance creating. In our opinion this is not needed. Let us take a step back and see what Theorem 2.3 proves. For any algorithm, any $\epsilon, \delta$ we define the error probability of the algorithm as $\Pr[error] = \Pr[\text{accuracy is }  \omega(\epsilon)] $. The theorem simply says that there cannot exist an algorithm with an error probability guarantee of $O(\max(p_\ell(\epsilon, \delta), p_{\ell'}(\epsilon, \delta))  - \epsilon)$; that simple. The proof of this theorem is through the construction of an instance, in which we show that the guarantee of  $O(\max(p_\ell(\epsilon, \delta), p_{\ell'}(\epsilon, \delta))  - \epsilon)$ is unattainable. Therefore, if for this instance the guarantee is unattainable, no algorithm comes with such a guarantee; this is precisely the statement of the theorem.
> > > >
> > > > The reviewer had reservations with respect to the introduction of the quantities $\max{p_\ell(\epsilon, \delta)}$. For one thing, you cannot deny the lower bound of Theorem 2.3 (the reviewer acknowledged its technical correctness). This theorem implies that the $p_\ell$ quantities provide a lower bound for the error probability! **This shows how important they are**. It shows that these values capture the inherent difficulty of an instance. And intuitively this makes sense; **The more dense the instance the harder it is to learn in it** In addition, combining Theorem 2.3 and 2.4 we indeed show almost tight results. We do not understand how the reviewer denies and discredits that.
> > > >
> > > > The reviewer wrongfully claims that our algorithmic upper bound is weak. Your main argument here is that $\mathcal{A} + \mathcal{B}$ tends to 0 with infinite samples. **This is so wrong**. That limit depends on the sparsity of the instance. As we mentioned in our previous reply, if the instance is sparse (elements are at distance $>3\epsilon$ from each other), the limit tends to 1!!! What does that show us again? It shows that sparsity is what need to account for. What captures sparsity in our case? The values $p_\ell$. Any tighter upper bound here would necessarily require assumptions on the density/sparsity (what you mention makes explicit assumptions. You alone talked about compact instances).

---

> > > > > ### Comment · Reviewer_qRef · 2023-08-15
> > > > >
> > > > > Theorem 2.3 is indeed incorrectly stated - it does not state what you are saying in the response.
> > > > >
> > > > > $\mathcal{A}+\mathcal{B}$ does indeed limit to zero when the space is compact and I cannot think of any realistic situation in which the space cannot be written as a subspace of a compact space. Note that I do not require each point $x$ to have another point within distance $\epsilon$ from it since the point $x$ itself is within this distance.
> > > > >
> > > > > As to your previous reply…
> > > > >
> > > > > What you have written in Theorem 2.3 is like writing that “it takes at least $L^2/4$ clock cycles to search for a number in an array of length $L$ because when $L=2$ we need at least $1=L^2/4$ clock cycles”. When somebody writes a lower bound in a paper, that lower bound has a certain “strength” which is written in the theorem statement - the less stuff that you can control in the proof the stronger that lower bound is. In your proof you can control absolutely everything which makes it so weak that nobody would ever put it in a paper (when trying to show an algorithm is almost optimal). In most machine learning problems there are instances which are completely unlearnable - using your logic you can use such an unlearnable instance to form any lower bound you like.
> > > > >
> > > > > I will not be increasing my score.

---

> > > > > > ### Author Response · Authors · 2023-08-15
> > > > > > **Reply to last comment and pointers to AC/meta-reviewer**
> > > > > >
> > > > > > *Theorem 2.3 is indeed incorrectly stated - it does not state what you are saying in the response*
> > > > > >
> > > > > > So let's say that we change the statement according to your feedback, and write that there exist instances in which the error probability of any algorithm cannot be $O(\max(p_\ell(\epsilon, \delta), p_{\ell'}(\epsilon, \delta)) - \epsilon)$. What does this imply? It implies that no algorithm can guarantee an error probability $O(\max(p_\ell(\epsilon, \delta), p_{\ell'}(\epsilon, \delta)) - \epsilon)$ in general. We do not understand the confusion here.
> > > > > >
> > > > > > *$\mathcal{A} + \mathcal{B}$ does indeed limit to zero when the space is compact and I cannot think of any realistic situation in which the space cannot be written as a subspace of a compact space*
> > > > > >
> > > > > > Do you understand that you are **EXPLICITLY** making assumptions here? You assume a compact space and you are talking about "realistic scenarios". We give bounds that hold for every input instance.
> > > > > >
> > > > > > *Note that I do not require each point x to have another point within distance $\epsilon$ from it since the point
> > > > > >  itself is within this distance.*
> > > > > >
> > > > > > In the sparse example we gave you in the previous comment, you need to have every point in your sample set if you are to make $\mathcal{A}$ and $\mathcal{B}$ 0. So the insight from this sort of analysis is the following: If you sample the whole support of $\mathcal{D}$ you have learned everything. That's the only conclusion your logic leads to and this conclusion is pretty meaningless. By the way, even if you take infinite samples, the support of the input distributions can be of a larger infinity, so in a sparse space $\mathcal{A}$ and $\mathcal{B}$ will be 1. In the meaningful case where you don't have the whole support as the sample set, you can **never** provide an upper bound that tends to 0, because there are problematic instances. You just chose to ignore this in your analysis (how can be an algorithm that learns the sparse instance perfectly? even with infinite samples there will be inputs in which it will fail).
> > > > > >
> > > > > > Out of curiosity can you give us a lower bound for any problem which you think is strong and appropriate?
> > > > > >
> > > > > > * In your proof you can control absolutely everything which makes it so weak that nobody would ever put it in a paper (when trying to show an algorithm is almost optimal).*
> > > > > >
> > > > > > In our proof we just construct an input instance that demonstrates hardness of learning. Actually, we only control the input distributions. Our results even hold for any $\epsilon, \delta \in (0,1)$. **Showing hardness of computation through the construction of pathogenic instances is a such a classical technique in TCS (see all NP-hardness proofs for instance)** Again, high level sketch of this proof technique: "You claim you can always learn with guarantee $\rho$? Ok, then here's an instance in which learning with $\rho$ is impossible for any algorithm. Hence, nobody can learn with guarantee at most $\rho$"
> > > > > >
> > > > > > * In most machine learning problems there are instances which are completely unlearnable*
> > > > > >
> > > > > > Usually, the learnability of problem is expressed in terms of some quantity/measure, for example the VC dimension. Here we simply express learnability in terms of $p_\ell$.
> > > > > >
> > > > > > *I will not be increasing my score.*
> > > > > >
> > > > > > We do not reply to you because we want you to increase your score. We simply hope the AC/meta-reviewer reads this conversation and understands that your feedback is totally erroneous.
> > > > > >
> > > > > > Overall, it seems to us that you fail to understand two things (**pointers to AC/meta-reviewer**):
> > > > > >
> > > > > > 1) Your critique of our "weak" upper bound relies on you making extra assumptions on the problem. Under those assumptions (compact and realistic space, sample set being the whole distribution support) you clearly make the problem much easier. So yes you will get a tighter bound.
> > > > > >
> > > > > > 2) In your critique of our lower bound you seem to miss the importance of hard examples. It's like you're saying "You can solve k-center with an approximation ratio better than 2, because most instances are not pathogenic." Our reply to this: "Well, OK, but no approximation algorithm can guarantee a ratio better than 2. If that was the case then the pathogenic instances would also be solvable".

---

### Author Rebuttal · Authors · 2023-08-04

We thank all of our reviewers for their time and their thoughtful comments. We reply to each one individual below.

---

### Decision · Program_Chairs · 2023-09-21

**Decision:**

Accept (poster)

**Comment:**

The reviewers have wildly different appraisals of this paper. The two reviewers who want to accept the paper have written very short reviews. The most negative reviewer seems not to have the right background to assess the paper, and is overconfident in their negative assessment. The last reviewer only criticizes the presentation, and only mildly. To me, the paper looks very well motivated, with excellent presentation - the authors have taken pains to explain their ideas clearly - interesting, and important.